# CRISP: Curriculum inducing Primitive Informed Subgoal Prediction for Hierarchical Reinforcement Learning

## Abstract

Hierarchical reinforcement learning is a promising approach that uses temporal abstraction to solve complex long horizon problems. However, simultaneously learning a hierarchy of policies is unstable as it is challenging to train higher-level policy when the lower-level primitive is non-stationary. In this paper, we propose to generate a curriculum of achievable subgoals for evolving lower-level primitives using reinforcement learning and imitation learning. The lower level primitive periodically performs data relabeling on a handful of expert demonstrations using our primitive informed parsing. We provide expressions to bound the sub-optimality of our method and develop a practical algorithm for hierarchical reinforcement learning. Since our approach uses a handful of expert demonstrations, it is suitable for most robotic control tasks. Experimental results on complex maze navigation and robotic manipulation environments show that inducing hierarchical curriculum learning significantly improves sample efficiency, and results in better learning of goal conditioned policies in temporally extended tasks.

## 1 Introduction

Reinforcement learning (RL) algorithms have made significant progress in solving continuous control tasks like performing robotic arm manipulation (Levine et al., 2015; Vecerík et al., 2017) and learning dexterous manipulation (Rajeswaran et al., 2017). However, the success of RL algorithms on complex long horizon continuous tasks has been limited by issues like long term credit assignment and inefficient exploration (Nachum et al., 2019; Kulkarni et al., 2016), especially in sparse reward scenarios (Andrychowicz et al., 2017). Hierarchical reinforcement learning (HRL) (Dayan & Hinton, 1993; Sutton et al., 1999; Parr & Russell, 1998) promises the benefits of temporal abstraction and efficient exploration for solving tasks that require long term planning. In *goal-conditioned* hierarchical framework, the high-level policy predicts subgoals for lower primitive, which in turn performs primitive actions directly on the environment (Nachum et al., 2018; Vezhnevets et al., 2017; Levy et al., 2017). However, simultaneously learning multi-level policies has been found to be challenging in practice due to non-stationary higher level state transition and reward functions.

Prior works have leveraged expert demonstrations to bootstrap learning (Nair et al., 2017; Rajeswaran et al., 2017; Hester et al., 2017). Some approaches rely on leveraging expert demonstrations via fixed parsing, and consequently bootstrapping multi-level hierarchical RL policy using imitation learning (Gupta et al., 2019). Generating an efficient subgoal transition dataset is *crucial* in such tasks. In this work, we propose an adaptive parsing technique for leveraging expert demonstrations and show that it outperforms fixed parsing based approaches on tasks that require long term planning. Ideally, a good subgoal should properly balance the task split between the hierarchical levels according to current goal reaching ability of the lower primitive, thus avoiding degenerate solutions. As the lower primitive improves, the subgoals provided to lower primitive should become progressively more difficult, such that *(i)* the subgoals are always achievable by the current lower level primitive, *(ii)* task split is properly balanced between hierarchical levels, and *(iii)* reasonable progress is made towards achieving the final goal. In this work, we introduce hierarchical curriculum learning to deal with non-stationarity issue. We build upon these ideas and propose a generally applicable HRL approach: *Curriculum inducing primitive informed subgoal prediction* (CRISP).

CRISP parses a handful of expert demonstrations using our novel subgoal relabeling method: primitive informed parsing (PIP). In PIP, current lower primitive is used to perform data relabeling on expert demonstrations dataset to generate effcent subgoal supervision for the higher level policy. Since the lower primitive performs data relabeling, this approach does not require explicit labeling or segmentation of demonstrations by an expert. The periodically generated higher level subgoal dataset is used with an additional imitation learning (IL) objective to provide curriculum based regularization for the higher policy. For imitation learning, we devise inverse reinforcement learning regularizer (Ghasemipour et al., 2020; Kostrikov et al., 2018; Ho & Ermon, 2016), which constraints the state marginal of the learned policy to be similar to that of the expert demonstrations. The details of CRISP, PIP, and IRL objective are mentioned in Section 3. We also derive sub-optimality bounds in Section 3.2 to theoretically justify the benefits of curriculum learning in hierarchical framework. Finally, we provide a practical approach to perform hierarchical reinforcement learning.

Since our approach uses a handful of expert demonstrations, it is generally applicable on most complex long horizon tasks. We perform experiments on random maze navigation and complex robotic pick and place environments, and empirically verify that the proposed approach clearly outperforms the baseline approaches on long horizon tasks.

## 2 BACKGROUND

We consider *Universal Markov Decision Process* (UMDP) (Schaul et al., 2015) setting, which is a Markov Decision process (MDP) augmented with the goal space $G$. UMDPs are represented as a 6-tuple $(S, A, P, R, \gamma, G)$, where $S$ is the state space, $A$ is the action space, $P(s^{'}|s, a) = \mathbb{P}(s_{t+1} = s^{'}|s_t = s, a_t = a)$ is the transition function that describes the probability of reaching state $s^{'}$ when the agent takes action $a$ in the current state $s$. The reward function $R$ generates rewards $r$ at every timestep, $\gamma$ is the discount factor, and $G$ is the goal space. In the UMDP setting, a fixed goal $g$ is selected for an episode, and $\pi(a|s, g)$ denotes the goal-conditioned policy. $d^{\pi}(s) = (1 - \gamma) \sum_{t=0}^{T} \gamma^t P(s_t = s|\pi)$ represents the discounted future state distribution, and $d_c^{\pi}(s) = (1 - \gamma^c) \sum_{t=0}^{T} \gamma^{tc} P(s_{tc} = s|\pi)$ represents the c-step future state distribution for policy $\pi$. The overall objective is to learn policy $\pi(a|s, g)$ which maximizes the expected future discounted reward objective $J = (1 - \gamma)^{-1} \mathbb{E}_{s \sim d^{\pi}, a \sim \pi(a|s,g), g \sim G} [r(s_t, a_t, g)]$

Let $s$ be the current state and $g$ be the final goal for the current episode. In our goal-conditioned hierarchical RL setup, the overall policy $\pi$ is divided into multi-level policies. The higher level policy $\pi^H(s_g|s, g)$ predicts subgoals (Dayan & Hinton, 1993) $s_g$ for the lower level primitive $\pi^L(a|s, s_g)$, which in turn executes primitive actions $a$ directly on the environment. The lower primitive $\pi^L$ tries to achieve subgoal $s_g$ within $c$ timesteps by maximizing intrinsic rewards $r_{in}$ provided by the higher level policy. The higher level policy $\pi^H$ gets extrinsic reward $r_{ex}$ from the environment, and predicts the next subgoal $s_g$ for the lower primitive. The process is continued until either the final goal $g$ is achieved, or the episode terminates. We consider sparse reward setting where the lower primitive is sparsely rewarded intrinsic reward $r_{in}$ if the agent reaches within $\delta^L$ distance of the predicted subgoal $s_g$: $r_{in} = 1(\|s_t - s_g\|_2 \leq \delta^L)$, and the higher level policy is sparsely rewarded extrinsic reward $r_{ex}$ if the achieved goal is within $\delta^H$ distance of the final goal $g$: $r_{ex} = 1(\|s_t - g\|_2 \leq \delta^H)$. We assume access to a handful of expert demonstrations $D = \{e^i\}_{i=1}^N$, where $e^i = (s_0^e, s_1^e, \ldots, s_{T-1}^e)$. We only assume access to demonstration states $s_i^e$ (and not demonstration actions) which can be obtained in most robotic control tasks.

## 3 METHODOLOGY

In this section, we explain our hierarchical curriculum learning based approach CRISP. An overview of the method is depicted in Figure 1. First, we formulate our primitive informed parsing method PIP, which periodically performs data relabeling on expert demonstrations to populate subgoal transition dataset. Then, we explain how we use this dataset to learn high level policy using reinforcement learning and additional inverse reinforcement learning(IRL) based regularization objective.

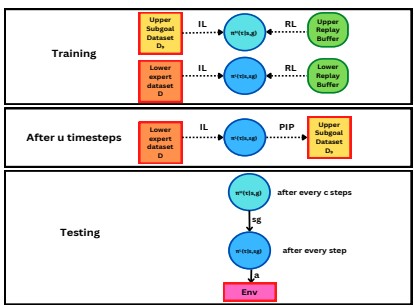

Figure 1: Overview of CRISP: While training, both upper policy($\pi^H$) and lower primitive($\pi^L$) are trained using RL and IL(imitation learning). After every u timesteps, $D_g$ is re-populated using PIP.

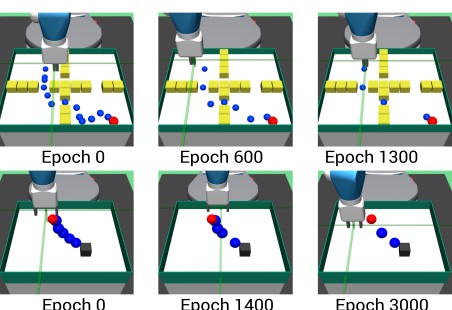

Figure 2: This figure depicts CRISP subgoals progress vs training epochs elapsed. The higher level policy generates a curriculum of subgoals for lower primitive(walls:yellow, final goal:red, subgoals:blue)

## 3.1 PRIMITIVE INFORMED PARSING: PIP

Primitive informed parsing approach uses the current lower primitive $\pi_L$ to parse expert state demonstrations dataset $\mathcal{D}$(PIP only requires expert state demonstrations and does not require expert actions from demonstrations, as explained later in this section.) The underlying idea is: PIP should select sequences of maximally temporally separated states from demonstration trajectory $e$. These maximally temporally separated state sequences constitute the higher level subgoal dataset $\mathcal{D}_g$. We explain below how PIP adaptively parses expert demonstration trajectories from $D$.

We start with current lower primitive $\pi_L$ and an expert state demonstration trajectory $e = (s_0^e, s_1^e, \ldots, s_{T-1}^e)$ . The environment is reset to state $s_0^e$. Starting at $i = 1$ to $T - 1$, we incrementally provide states $s_i^e$ as subgoals to lower primitive $\pi_L$. From $s_0^e$, $\pi_L$ tries to achieve $s_i^e$ within $c$ timesteps. If $\pi_L$ fails to achieve the subgoal $s_i^e$ from the initial state, we add $s_i^e$ to the list of subgoals. The underlying idea is that since $s_{i-1}^e$ was the last subgoal achieved by lower-level primitive, it thus makes a good candidate for maximally reachable subgoal. Once we have added $s_{i-1}^e$ to the list of subgoals, we continue the process after setting $s_{i-1}^e$ as the new initial state until we reach the end of demonstration trajectory $e$. This subgoal transition sequence thus collected is added to $\mathcal{D}_g$. The method thus populates the subgoal transition dataset with maximally temporally separated achievable subgoals.

---

**Algorithm 1** PIP: Primitive Informed Parsing

---

1: Initialize $D_g = \{\}$
2: **for** each trajectory $e = (s_0^e, s_1^e, \ldots, s_{T-1}^e)$ in $\mathcal{D}$ **do**
3:     initial state $\leftarrow s_0^e$
4:     final goal $\leftarrow g$
5:     Initialize list of subgoals $D_g^e = \{\}$
6:     **for** i = 1 **to** $T - 1$ **do**
7:         Reset to initial state
8:         Pass $s_i^e$ as the current goal to $\pi_L$
9:         **if** $s_i^e$ is not achieved by $\pi_L$ in $c$ time-steps **then**
10:            Add $\left(\text{initial state}, s_{i)}^e, \text{final goal}\right)$ to $D_g^e$
11:            initial state $\leftarrow s_{i-1}^e$
12:     $D_g \leftarrow D_g \cup D_g^e$

---

The pseudocode for PIP is given in Algorithm 1. PIP maximizes the utility of the lower level primitive while ensuring that it always receives achievable subgoals. However, it assumes that we can reset the environment to any state in $\mathcal{D}$ while collecting subgoal dataset. We discuss different ways to relax this assumption in Section 6.

## 3.2 SUBOPTIMALITY ANALYSIS

In this section, we analyze the suboptimality of our method, and examine how the performance benefits from curriculum learning and imitation learning objective. Let $\pi^*$ and $\pi^{**}$ be the unknown higher level and lower level optimal policies respectively, $\pi_{\theta_H}^H$ be our high level CRISP policy, and $\pi_{\theta_L}^L$ be our lower CRISP primitive policy, where $\theta_H$ and $\theta_L$ are trainable parameters of higher and lower level policies respectively. $D_{TV}(\pi_1, \pi_2)$ denotes total variation divergence between probability distributions $\pi_1$ and $\pi_2$. $s$ is the current state, $g$ is the final episodic goal, $s_g$ is the subgoal provided by upper level policy and $\tau$ are $c$ length sub-trajectories. Let $\Pi_D^H$ and $\Pi_D^L$ be the upper level probability distributions which generate datasets $D_H$ and $D_L$ respectively, $\kappa$ is some distribution over states and actions, and $G$ is the goal space. Firstly, we extend the definition from (Ajay et al., 2020) to goal-conditioned policies:

**Definition 1.** $\pi^*$ is $\phi_D$-common in $\Pi_D^H$, if $\mathbb{E}_{s \sim \kappa, \pi_D^H \sim \Pi_D^H, g \sim G}[D_{TV}(\pi^*(\tau|s,g)||\pi_D^H(\tau|s,g))] \leq \phi_D$

We define the suboptimality of policy $\pi$ with respect to optimal policy $\pi^*$ as:

$$Subopt(\theta) = |J(\pi^*) - J(\pi)| \tag{1}$$

**Theorem 1.** Assuming the optimal policy $\pi^*$ is $\phi_D$ common in $\Pi_D^H$, the suboptimality of upper policy $\pi_{\theta_H}^H$, over $c$ length sub-trajectories $\tau$ sampled from $d_c^{\pi^*}$ can be bounded as:

$$|J(\pi^*) - J(\pi_{\theta_H}^H)| \leq \lambda_H * \phi_D + \lambda_H * \mathbb{E}_{s \sim \kappa, \pi_D^H \sim \Pi_D^H, g \sim G}[D_{TV}(\pi_D^H(\tau|s,g)||\pi_{\theta_H}^H(\tau|s,g))]] \tag{2}$$

where $\lambda_H = \frac{2}{(1-\gamma)(1-\gamma^c)}R_{max}\|\frac{d_c^{\pi^*}}{\kappa}\|_\infty$

Furthermore, the suboptimality of lower primitive $\pi_{\theta_L}^L$ can be bounded as:

$$|J(\pi^{**}) - J(\pi_{\theta_L}^L)| \leq \lambda_L * \phi_D + \lambda_L * \mathbb{E}_{s \sim \kappa, \pi_D^L \sim \Pi_D^L, s_g \sim \pi_{\theta_L}^L}[D_{TV}(\pi_D^L(\tau|s,s_g)||\pi_{\theta_L}^L(\tau|s,s_g))]] \tag{3}$$

where $\lambda_L = \frac{2}{(1-\gamma)^2}R_{max}\|\frac{d_c^{\pi^{**}}}{\kappa}\|_\infty$

The proofs for Equations 2 and 3 are provided in Appendix A.1 and A.2 respectively. Equation 2 can be rearranged to yield the following form:

$$J(\pi^*) \geq J(\pi_{\theta_H}^H) - \lambda_H * \phi_D - \lambda_H * \mathbb{E}_{s \sim \kappa, \pi_D^H \sim \Pi_D^H, g \sim G}[d(\pi_D^H(\tau|s,g)||\pi_{\theta_H}^H(\tau|s,g))] \tag{4}$$

where, $d(\pi_D^H(\tau|s,g)||\pi_{\theta_H}^H(\tau|s,g)) = D_{TV}(\pi_D^H(\tau|s,g)||\pi_{\theta_H}^H(\tau|s,g))$

This can be solved as a minorize maximize algorithm which intuitively means: the overall objective can be optimized by $(i)$ maximizing the objective $J(\pi_{\theta_H}^H)$ via RL, and $(ii)$ minimizing TV divergence between $\pi^*$ and $J(\pi_{\theta_H}^H)$. We use entropy regularized RL technique Soft Actor Critic (Haarnoja et al., 2018a) to maximize $J(\pi_{\theta_H}^H)$.

## 3.3 HIERARCHICAL CURRICULUM LEARNING

In Equation 2, the suboptimality bound is dependent on $\phi_{D_g}$, which represents how good is the subgoal dataset $D_g$ populated by PIP. A lower value of $\phi_{D_g}$ implies that the optimal policy $\pi^*$ is closely represented by the dataset $D_g$. Since we use lower primitive to parse expert demonstrations, as the lower primitive gets better, $\pi_{D_g}$ gets closer to $\pi^*$. Hence $D_g$ improves and the value of parameter $\phi_D$ decreases, which implies that suboptimality bound in Equation 2 gets tighter.

To implement curriculum learning while generating subgoal transition dataset $D_g$, the dataset $D_g$ is cleared after every $u$ timesteps, and re-populated using PIP, as explained in Algorithm 2. Periodically re-populating buffer $D_g$ after $u$ timesteps generates a natural curriculum for lower primitive, as shown in Figure 2.

### 3.4 IMITATION LEARNING REGULARIZATION FOR HIGHER-LEVEL POLICY

Different approximations to the distance function $d$ in Equation 4 yield different imitation learning regularizers. If $d$ is replaced by Kullback–Leibler divergence, the imitation learning regularizer becomes behavior cloning objective(BC) (Nair et al., 2017). If we replace $d$ with Jensen-Shannon divergence, the imitation learning objective takes the form of Inverse reinforcement learning(IRL) objective. Henceforth, CRISP-IRL will denote our method CRISP with IRL regularizer, and CRISP-BC will denote our method CRISP with BC regularizer. We consider both behavior cloning and IRL objectives in our experiments. We devise IRL objective as a GAIL (Ho & Ermon, 2016) like objective implemented using LSGAN (Mao et al., 2016). Let $(s^e, g^e, s_g^e) \sim D_g$ be a subgoal transition where $s^e$ is a state in an expert trajectory, $g^e$ is the corresponding final goal and $s_g^e$ is the subgoal. Let $s_g$ be the subgoal predicted by the high level policy $\pi_\theta^H(\cdot|s^e, g^e)$ and $\mathbb{D}_\epsilon^H$ be the higher level discriminator with parameters $\epsilon$. We bootstrap the learning of higher level policy by optimizing:

$$\max_{\pi_\theta^H} \min_\epsilon \frac{1}{2} \mathbb{E}_{(s^e, g^e, s_g^e) \sim D_g} [\mathbb{D}_\epsilon^H(s_g^e) - 1]^2 + \frac{1}{2} \mathbb{E}_{(s^e, g^e) \sim D_g, s_g \sim \pi_\theta^H(\cdot|s^e, g^e)} [\mathbb{D}_\epsilon^H(\pi_\theta^H(\cdot|s^e, g^e)) - 0]^2$$

(5)

This objective forces the higher policy subgoal predictions to be close to subgoal predictions of the dataset $\mathcal{D}_g$. For brevity, let $J_D^H$ and $J_D^L$ represent upper and lower IRL objectives, which depend on parameters $(\theta_H, \epsilon_H)$ and $(\theta_L, \epsilon_L)$ respectively. The discriminator $\mathbb{D}_\epsilon^H$ creates a natural curriculum for regularizing higher level policy by assigning the value 1 to the predicted subgoals that are closer to the subgoals from dataset $D_g$, and 0 otherwise. The discriminator improves with training, and regularizes the higher policy to predict achievable subgoals.

### 3.5 POLICY OPTIMIZATION

The higher level policy is trained to produce subgoals, which when fed into the lower level primitive, maximize the sum of future discounted rewards for our task using off-policy reinforcement learning. Here, $\pi_\theta^L$ is the current lower primitive, $s_t$ is the state at time $t$, $T$ is the task horizon and $g$ is the sampled goal for the current episode. For brevity, we can refer to this objective function as $J_{\theta_H}^H$ and $J_{\theta_L}^L$ for upper and lower policies. We use the IRL objective to leverage the primitive-parsed dataset $\mathcal{D}_g$. Therefore, the high level policy is trained by optimizing

$$\max_{\theta_H} J_{\theta_H}^H + \psi(\min_{\epsilon_H} J_D^H(\theta_H, \epsilon_H))$$

(6)

Whereas, the lower level primitive is trained by optimizing,

$$\max_{\theta_L} J_{\theta_L}^L + \psi(\min_{\epsilon_L} J_D^L(\theta_L, \epsilon_L))$$

(7)

---

**Algorithm 2** CRISP

---

**Require:** Expert state demonstrations $D$, hyperparameter $u$
 1: Initialize higher level subgoal transition dataset $D_g = \{\}$
 2: **for** epoch $i = 1 \ldots N$ **do**
 3:     **if** $i \% u == 0$ **then**
 4:         Clear $D_g$
 5:         Populate $D_g$ by relabeling $D$ using PIP
 6:     **for** $j = 1$ **to** $T - 1$ **do**
 7:         Collect off policy experience using $\pi_H$ and $\pi_L$
 8:     Update lower primitive via Soft Actor Critic (SAC) and IRL(Eq 7)
 9:     Sample transitions from $D_g$
10:     Update higher policy via SAC and IRL(Eq 6)

---

The lower policy is regularized using primitive expert demonstration dataset, and the upper level is optimized using subgoal transition dataset populated using PIP. $\psi$ is the regularization weight hyperparameter for the IRL objective. When $\psi = 0$, the method reduces to HRL policy with no higher level policy regularization. When $\psi$ is too high, the method might overfit to the expert demonstration dataset. We perform ablation analysis to choose $\psi$ in our experiments in Appendix A.3. The CRISP algorithm is shown in Algorithm 2

## 4 RELATED WORK

Learning effective hierarchies of policies has garnered substantial research interest in RL (Barto & Mahadevan, 2003; Sutton et al., 1999; Parr & Russell, 1998; Dietterich, 1999). Options framework (Sutton et al., 1999; Bacon et al., 2016; Harutyunyan et al., 2017; Harb et al., 2017; Harutyunyan et al., 2019; Klissarov et al., 2017) learns temporally extended macro actions, and a termination function for solving long horizon tasks. However, these approaches run into degenerate solutions in absence of proper regularization, where a sub-policy either terminates after each step, or runs for the entire episode. In goal-conditioned learning, some previous approaches restrict the search space by greedily solving for specific goals (Kaelbling, 1993; Foster & Dayan, 2002). This approach has also been extended to hierarchical RL (Wulfmeier et al., 2019; 2020; Ding et al., 2019). HIRO (Nachum et al., 2018) and HRL with hindsight (Levy et al., 2017) approaches deal with the non-stationarity issue in hierarchical learning by relabeling transition data for training goal-conditioned policies, where the higher level predicts subgoals for the lower primitive. In contrast, our method deals with non-stationarity by regularizing the higher policy with imitation learning to provide a curriculum of achievable subgoals to the lower primitive. Our approach is inspired from curriculum learning (Bengio et al., 2009), where the task difficulty for the lower primitive gradually increases in complexity, thereby amortizing non-stationarity.

Previous approaches that leverage expert demonstrations have shown impressive results (Nair et al., 2017; Rajeswaran et al., 2017; Hester et al., 2017). Expert demonstrations have also been used to bootstrap option learning (Krishnan et al., 2017b; Fox et al., 2017; Shankar & Gupta, 2020; Kipf et al., 2019). Other approaches use imitation learning to bootstrap hierarchical approaches in complex task domains (Shiarlis et al., 2018; Krishnan et al., 2017a; 2019; Kipf et al., 2019). Relay Policy Learning (RPL) (Gupta et al., 2019) parses uses simple fixed window based approach for parsing expert demonstrations to generate subgoal transition dataset for training higher level policy. However, fixed parsing based approaches might predict subgoals that are either too hard for the lower level primitive, in which case the higher level is cursed with ambiguous extrinsic reward signal, or too easy subgoals, in which case the higher level is forced to do most of the heavy-lifting for solving the task. In contrast, our data relabeling technique PIP segments expert demonstration trajectories into *meaningful* subtasks, without requiring an external expert. Our adaptive parsing approach considers the limited goal reaching ability of lower primitive, and is therefore able to produce much better subgoals.

## 5 EXPERIMENTS

For experimental analysis, we considered various complex tasks with continuous state and action spaces. We perform experiments on the following two robotic Mujoco (Todorov et al., 2012) environments: $(i)$ maze navigation, and $(ii)$ pick and place environment. These environments employ a 7-DoF robotic arm to perform maze navigation and robotic manipulation. We empirically show the performance comparison of our approach with various baselines in challenging task domains.

### 5.1 COMPARATIVE ANALYSIS

Here, we enlist the baseline methods for comparison, and explain the rationale.

- **Relay Policy Learning (RPL)** (Gupta et al., 2019) parses subgoals from expert state demonstrations using a fixed window approach. We use this baseline to highlight the advantage of adaptive parsing of subgoals compared to fixed window parsing. We perform extensive search for the window size hyper-parameters in RPL for each environment, which we provide in appendix A.

- **Hierarchical** 2-**level policy (Hier)** denotes a hierarchical policy where the high level policy is trained using only reinforcement learning, and the lower level policy is trained using RL and IRL using primitive expert demonstrations. We use it to show the importance of curriculum based subgoal generation and consequent IRL based regularization on the higher level policy. The hierarchical 2-level policy where both the upper and lower levels are trained using only RL failed to show good performance compared to other methods. Hence, we do not include it in the baseline comparisons.

- **Discriminator Actor Critic (DAC)** (Kostrikov et al., 2018) uses IRL to learn a single-level policy using low level expert demonstrations $\mathcal{D}$. Using this baseline, we demonstrate the advantage of using hierarchy, curriculum based subgoal generation and consequent IRL based regularization in our approach.

## 5.2 ROBOTIC MAZE NAVIGATION ENVIRONMENT

Here we provide details about the maze navigation environment, its implementation and results.

### 5.2.1 ENVIRONMENT SETUP

In this environment, a 7-DOF robotic arm gripper navigates across random four room mazes. The gripper arm is kept closed and the positions of walls and gates are randomly generated. The table is discretized into a rectangular $W * H$ grid, and the vertical and horizontal wall positions $W_P$ and $H_P$ are randomly picked from $(1, W - 2)$ and $(1, H - 2)$ respectively. In the four room environment thus constructed, the four gate positions are randomly picked from $(1, W_P - 1)$, $(W_P + 1, W - 2)$, $(1, H_P - 1)$ and $(H_P + 1, H - 2)$. The height of gripper is kept fixed at table height, and it has to navigate across the maze to the goal position(shown as red sphere). The maximum task horizon $T$ is kept at 225 timesteps, and the lower primitive is allowed to execute for $c = 15$ timesteps.

### 5.2.2 IMPLEMENTATION DETAILS

The following implementation details refer to both the higher and lower level polices, unless otherwise explicitly stated. The state and action spaces in the environment are continuous. The actor, critic and discriminator networks are formulated as 3 layer fully connected neural networks with 512 neurons in each layer. The state is represented as the vector $[p, \mathcal{M}]$, where $p$ is current gripper position and $\mathcal{M}$ is the sparse maze array. The higher level policy input is thus a concatenated vector $[p, \mathcal{M}, g]$, where $g$ is the target goal position, whereas the lower level policy input is concatenated vector $[p, \mathcal{M}, s_g]$, where $s_g$ is the sub-goal provided by the higher level policy. The current position of the gripper is the current achieved goal. The sparse maze array $\mathcal{M}$ is a discrete $2D$ one-hot vector array, where $1$ represents presence of a wall block, and $0$ absence. In our experiments, the size of $p$ and $\mathcal{M}$ are kept to be 3 and 110 respectively. The upper level predicts subgoal $s_g$, hence the higher level policy action space dimension is the same as the dimension of goal space. The lower primitive action $a$ which is directly executed on the environment, is a $4$ dimensional vector with every dimension $a_i \in [0, 1]$. The first 3 dimensions provide offsets to be scaled and added to gripper position for moving it to the intended position. The last dimension provides gripper control(0 implies a fully closed gripper, 0.5 implies a half closed gripper and 1 implies a fully open gripper). We select 100 randomly generated mazes each for training, testing and validation. For selecting train, test and validation mazes, we first randomly generate 300 distinct mazes, and then randomly divide them into 100 train, test and validation mazes each. Each experiment is run on 4 parallel workers. We use off-policy Soft Actor Critic (Haarnoja et al., 2018b) algorithm for optimizing RL objective in our experiments. We keep the regularization weight hyperparameter as $\Psi = 0.0078$ in our experiments. We use Adam (Kingma & Ba, 2014) optimizer in our experiments. The hyperparameter $u$ which is the number of training iterations after which the replay buffer is flushed and re-populated is set as 100. The experiments are run for $2.93e6$ timesteps. The method for generating expert demonstrations is provided in Appendix A.4.

### 5.2.3 RESULTS

In Table 1, we report the success rate performance of the proposed methods and other baselines in the maze navigation and pick and place environments averaged over 3 seeds. CRISP-IRL is CRISP with Inverse RL(IRL) regularization objective, and CRISP-BC is CRISP with behavior cloning(BC) regularization. While training and testing, we evaluate success rates over $N = 100$ random episodic rollouts. Since the test mazes are randomly generated, the performance also proves a measure of generalization capability of our proposed approach. From the results, it is evident that the proposed approach outperforms the baselines, and demonstrates impressive generalization.

Table 1: Success rates: Maze navigation and Pick and place environment

| Method | Maze Navigation | Pick and Place |
|---|---|---|
| CRISP-IRL | **$0.65 \pm 0.07$** | **$0.88 \pm 0.03$** |
| CRISP-BC | $0.53 \pm 0.02$ | **$0.73 \pm 0.01$** |
| RPL | $0.57 \pm 0.035$ | $0.02 \pm 0.01$ |
| Hier | $0.52 \pm 0.08$ | $0.02 \pm 0.02$ |
| DAC | $0.27 \pm 0.06$ | $0.38 \pm 0.07$ |

## 5.3 ROBOTIC PICK AND PLACE ENVIRONMENT

Here we provide details about the pick and place environment, its implementation and results.

### 5.3.1 ENVIRONMENT SETUP

In this environment, a 7-DOF robotic arm gripper has to pick a square block and bring/place it to a goal position. We set the goal position slightly higher than table height. The maximum task horizon $T$ is kept at 225 timesteps, and the lower primitive is allowed to execute for $c = 15$ timesteps. In this complex task, the gripper has to navigate to the block, close the gripper to hold the block, and then bring the block to the desired goal position. We provide the success rate results and baseline comparisons in section 5.3.3

### 5.3.2 IMPLEMENTATION DETAILS

In this environment, the actor, critic, and discriminator networks are formulated as 3 layer fully connected networks with 512 neurons in each layer. The state is represented as the vector $[p, o, q, e]$, where $p$ is current gripper position, $o$ is the position of the block object placed on the table, $q$ is the relative position of the block with respect to the gripper, and $e$ consists of linear and angular velocities of the gripper and the block object. The higher level policy input is thus a concatenated vector $[p, o, q, e, g]$, where $g$ is the target goal position. The lower level policy input is concatenated vector $[p, o, q, e, s_g]$, where $s_g$ is the sub-goal provided by the higher level policy. The current position of the block object is the current achieved goal. In our experiments, the sizes of $p$, $o$, $q$, $e$ are kept to be 3, 3, 3 and 11 respectively. The upper level predicts subgoal $s_g$, hence the higher level policy action space and goal space have the same dimension. The lower primitive action $a$ is a 4 dimensional vector with every dimension $a_i \in [0, 1]$. The first 3 dimensions provide gripper position offsets, and the last dimension provides gripper control(0 means closed gripper and 1 means open gripper). While training, the position of block object and goal are randomly generated(block is always initialized on the table, and goal is always above the table at a fixed height). We select 100 randomly pick and place environments each for training, testing and validation. For selecting train, test and validation mazes, we first randomly generate 300 distinct environments with different block and target goal positions, and then randomly divide them into 100 train, test and validation mazes each. Each experiment is run on 4 parallel workers. We use off-policy Soft Actor Critic (Haarnoja et al., 2018b) algorithm for the RL objective in our experiments. We keep the regularization weight hyperparameter as $\Psi = 0.005$ in our experiments, and use Adam (Kingma & Ba, 2014) optimizer in our experiments. The hyperparameter $u$ which is the number of training iterations after which the replay buffer is flushed and re-populated is set as 50. The experiments are run for $6.75e6$ timesteps. The method for generating expert demonstrations is provided in Appendix A.5.

### 5.3.3 RESULTS

In Table 1, we report the success rate performances averaged over 3 seeds. CRISP-IRL is CRISP with Inverse RL(IRL) regularization, and CRISP-BC is CRISP with behavior cloning(BC) regularization. While training and testing, we evaluate success rates over $N = 100$ random episodic rollouts. From Table 1 it is apparent that CRISP-BC and CRISP-IRL clearly outperform the baselines by a large margin. This provides convincing evidence that stable hierarchical learning indeed demonstrates better performance on complex long horizon tasks.

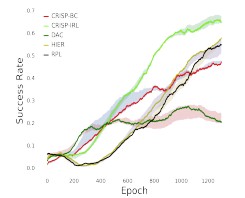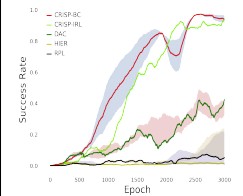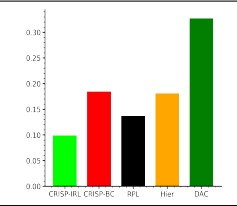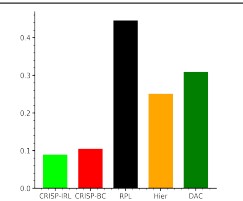

Figure 3: The success rate plots show performance comparison between our method and baselines in maze navigation (Column 1) and pick and place (Column 2) versus number of training epochs. Columns 3 and 4 plots compare the methods via distance metric(average distance between achieved goal and final goal in 100 episodic rollouts) in room maze navigation (Column 3) and pick and place (Column 4). As can be seen, our methods show impressive performance compared to baselines.

## 5.4 ABLATIVE STUDIES

To elucidate the importance of various constituent design choices in our proposed approach, we show success rate comparison plots in Figure 3 columns 1 and 2. We compare our proposed approach with *RPL* to demonstrate the advantage of adaptive parsing over fixed window parsing and thus segmenting the task into meaningful subtasks using the lower primitive, while using curriculum of subgoals for evolving lower primitive. The comparison with *Hier* method shows the advantage of curriculum based subgoal regularization using imitation learning. Finally, *DAC* highlights the importance of using hierarchy and curriculum based subgoal regularization using imitation learning. CRISP shows faster convergence and stable learning when compared to other approaches. In Figure 3 columns 3 and 4, we compare the methods in terms of distance between final achieved goal and desired goal averaged over 100 episodic rollouts. This metric gives an idea of how accurately an approach solves the task. CRISP clearly demonstrates better accuracy in reaching the final goal.

The number of expert demonstrations is kept to be 100 after performing ablation experiments as shown in Figure 4 in Appendix A.3. If the number of demonstrations is less, the policy might overfit to the demonstrations, thereby hampering overall performance, as depicted in Figure 4. Although the number of available expert demonstrations is generally dependent on the task environment, we increased the expert demonstrations until there was no significant performance improvement. It is important to note that since CRISP uses PIP to select good subgoals from lower level expert dataset, we require "good" lower level expert demonstration trajectories. If the expert trajectory is "bad", PIP is unable to select good subgoals, leading to poor performance.

We perform ablation experiments for choosing hyperparameter $u$, which is the number of training iterations after which the replay buffer is flushed and re-populated. The ablation experiments are provided in 5 in Appendix A.3. For RPL experiments, we choose the window size hyper-parameter $c$ by running RPL experiments for different values of $c$. The experiments are shown in Figure 7 in Appendix A.3. After performing experiments in maze navigation and pick and place environments, $c$ is set to 4 and 8 respectively. We also performed ablation analysis for choosing the imitation learning weight hyperparameter $\lambda$. The experiments are shown in Figure 6 in Appendix A.3.

## 6 DISCUSSION AND FUTURE WORK

We introduce CRISP, which is our general purpose lower level primitive informed method CRISP for efficient hierarchical reinforcement learning. CRISP leverages primitive parsed expert demonstrations and performs data relabeling on expert demonstrations to populate subgoal transition dataset for regularizing higher level policy. CRISP employs hierarchical curriculum learning to deal with non-stationarity. We evaluate our method on complex robotic maze navigation and pick and place manipulation tasks, and demonstrate that it makes substantial gains over its baselines.

However, CRISP assumes the ability to reset the environment to any state from expert demonstrations dataset while collecting subgoal dataset using PIP. A possible method for relaxing this assumption is to combine CRISP with (Eysenbach et al., 2017) that learns a backward controller that tries to reset the environment. This approach is an interesting avenue for future work.

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

# A    APPENDIX

## A.1    SUB-OPTIMALITY ANALYSIS PROOF FOR HIGHER LEVEL POLICY

The sub-optimality of upper policy $\pi_{\theta_H}^H$, over $c$ length sub-trajectories $\tau$ sampled from $d_c^{\pi^*}$ can be bounded as:

$$|J(\pi^*) - J(\pi_{\theta_H}^H)| \leq \lambda_H * \phi_D + \lambda_H * \mathbb{E}_{s \sim \kappa, \pi_D^H \sim \Pi_D^H, g \sim G}[D_{TV}(\pi_D^H(\tau|s,g)||\pi_{\theta_H}^H(\tau|s,g))]] \quad (8)$$

where $\lambda_H = \frac{2}{(1-\gamma)(1-\gamma^c)} R_{max} \|\frac{d_c^{\pi^*}}{\kappa}\|_\infty$

*Proof.* We extend the suboptimality bound from (Ajay et al., 2020) between goal conditioned policies $\pi^*$ and $\pi_{\theta_H}^H$ as follows:

$$|J(\pi^*) - J(\pi_{\theta_H}^H)| \leq \frac{2}{(1-\gamma)(1-\gamma^c)} R_{max} \mathbb{E}_{s \sim d_c^{\pi^*}, g \sim G}[D_{TV}(\pi^*(\tau|s,g)||\pi_{\theta_H}^H(\tau|s,g))] \quad (9)$$

By applying triangle inequality:

$$D_{TV}(\pi^*(\tau|s,g)||\pi_{\theta_H}^H(\tau|s,g)) \leq D_{TV}(\pi^*(\tau|s,g)||\pi_D^H(\tau|s,g)) + D_{TV}(\pi_D^H(\tau|s,g)||\pi_{\theta_H}^H(\tau|s,g)) \tag{10}$$

Taking expectation wrt $s \sim \kappa$, $g \sim G$ and $\pi_D^H \sim \Pi_D^H$,

$$\mathbb{E}_{s \sim \kappa, g \sim G}[D_{TV}(\pi^*(\tau|s,g)||\pi_{\theta_H}^H(\tau|s,g))] \leq \mathbb{E}_{s \sim \kappa, \pi_D^H \sim \Pi_D^H, g \sim G}[D_{TV}(\pi^*(\tau|s,g)||\pi_D^H(\tau|s,g))] +$$
$$\mathbb{E}_{s \sim \kappa, \pi_D^H \sim \Pi_D^H, g \sim G}[D_{TV}(\pi_D^H(\tau|s,g)||\pi_{\theta_H}^H(\tau|s,g))] \tag{11}$$

Since $\pi^*$ is $\phi_D$ common in $\Pi_D^H$, we can write 11 as:

$$\mathbb{E}_{s \sim \kappa, g \sim G}[D_{TV}(\pi^*(\tau|s,g)||\pi_{\theta_H}^H(\tau|s,g))] \leq \phi_D + \mathbb{E}_{s \sim \kappa, \pi_D^H \sim \Pi_D^H, g \sim G}[D_{TV}(\pi_D^H(\tau|s,g)||\pi_{\theta_H}^H(\tau|s,g))] \tag{12}$$

Substituting the result from Equation 12 in Equation 9, we get

$$|J(\pi^*) - J(\pi_{\theta_H}^H)| \leq \lambda_H * \phi_D + \lambda_H * \mathbb{E}_{s \sim \kappa, \pi_D^H \sim \Pi_D^H, g \sim G}[D_{TV}(\pi_D^H(\tau|s,g)||\pi_{\theta_H}^H(\tau|s,g))]] \quad (13)$$

where $\lambda_H = \frac{2}{(1-\gamma)(1-\gamma^c)} R_{max} \|\frac{d_c^{\pi^*}}{\kappa}\|_\infty$    □

## A.2    SUB-OPTIMALITY ANALYSIS PROOF FOR LOWER LEVEL POLICY

Let the optimal lower level policy be $\pi^{**}$. The suboptimality of lower primitive $\pi_{\theta_L}^L$ can be bounded as follows:

$$|J(\pi^{**}) - J(\pi_{\theta_L}^L)| \leq \lambda_L * \phi_D + \lambda_L * \mathbb{E}_{s \sim \kappa, \pi_D^L \sim \Pi_D^L, s_g \sim \pi_{\theta_L}^L}[D_{TV}(\pi_D^L(\tau|s,s_g)||\pi_{\theta_L}^L(\tau|s,s_g))]] \tag{14}$$

where $\lambda_L = \frac{2}{(1-\gamma)^2} R_{max} \|\frac{d_c^{\pi^{**}}}{\kappa}\|_\infty$

*Proof.* We extend the suboptimality bound from (Ajay et al., 2020) between goal conditioned policies $\pi^{**}$ and $\pi_{\theta_L}^L$ as follows:

$$|J(\pi^{**}) - J(\pi_{\theta_L}^L)| \leq \frac{2}{(1-\gamma)^2} R_{max} \mathbb{E}_{s \sim d_c^{\pi^{**}}, s_g \sim \pi_{\theta_L}^L}[D_{TV}(\pi^{**}(\tau|s,s_g)||\pi_{\theta_L}^L(\tau|s,s_g))] \quad (15)$$

By applying triangle inequality:

$$D_{TV}(\pi^{**}(\tau|s,s_g)||\pi_{\theta_L}^L(\tau|s,s_g)) \leq D_{TV}(\pi^{**}(\tau|s,s_g)||\pi_D^L(\tau|s,s_g)) + D_{TV}(\pi_D^L(\tau|s,s_g)||\pi_{\theta_L}^L(\tau|s,s_g)) \tag{16}$$

Taking expectation wrt $s \sim \kappa$, $s_g \sim \pi_{\theta_L}^L$ and $\pi_D^L \sim \Pi_D^L$,

$$\mathbb{E}_{s\sim\kappa,s_g\sim\pi_{\theta_L}^L}[D_{TV}(\pi^{**}(\tau|s,s_g)||\pi_{\theta_L}^L(\tau|s,s_g))] \leq \mathbb{E}_{s\sim\kappa,\pi_D^L\sim\Pi_D^L,s_g\sim\pi_{\theta_L}^L}[D_{TV}(\pi^{**}(\tau|s,s_g)||\pi_D^L(\tau|s,s_g))]+$$
$$\mathbb{E}_{s\sim\kappa,\pi_D^L\sim\Pi_D^L,s_g\sim\pi_{\theta_L}^L}[D_{TV}(\pi_D^L(\tau|s,s_g)||\pi_{\theta_L}^L(\tau|s,s_g))]$$
$$(17)$$

Since $\pi^{**}$ is $\phi_D$ common in $\Pi_D^L$, we can write 17 as:

$$\mathbb{E}_{s\sim\kappa,s_g\sim\pi_{\theta_L}^L}[D_{TV}(\pi^{**}(\tau|s,s_g)||\pi_{\theta_L}^L(\tau|s,s_g))] \leq \phi_D + \mathbb{E}_{s\sim\kappa,\pi_D^L\sim\Pi_D^L,s_g\sim\pi_{\theta_L}^L}[D_{TV}(\pi_D^L(\tau|s,s_g)||\pi_{\theta_L}^L(\tau|s,s_g))]$$
$$(18)$$

Substituting the result from Equation 18 in Equation 15, we get

$$|J(\pi^{**}) - J(\pi_{\theta_L}^L)| \leq \lambda_L * \phi_D + \lambda_L * \mathbb{E}_{s\sim\kappa,\pi_D^L\sim\Pi_D^L,s_g\sim\pi_{\theta_L}^L}[D_{TV}(\pi_D^L(\tau|s,s_g)||\pi_{\theta_L}^L(\tau|s,s_g))]]$$
$$(19)$$

where $\lambda_L = \frac{2}{(1-\gamma)^2}R_{max}\|\frac{d_c^{\pi^{**}}}{\kappa}\|_\infty$ $\hfill\square$

### A.3 ABLATION EXPERIMENTS

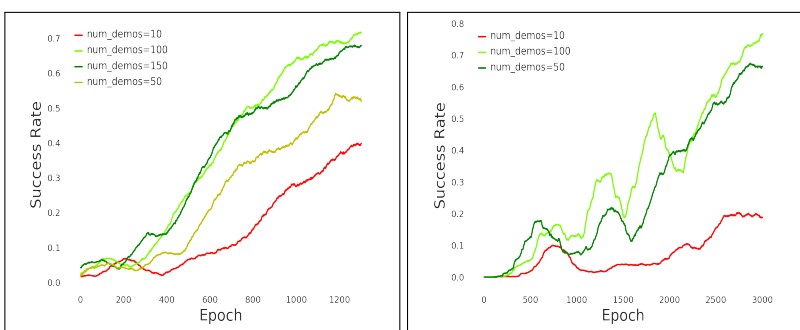

Figure 4: Column 1(maze navigation environment) and column 2(pick and place environment) show success rate performance plots of varying number of expert demonstrations versus number of training epochs. The number of expert demos should not be too less to cause the policy to overfit. We increase the number demonstrations until there is no significant progress in policy performance. Based on the experiments, we choose 100 expert demonstrations for both environments.

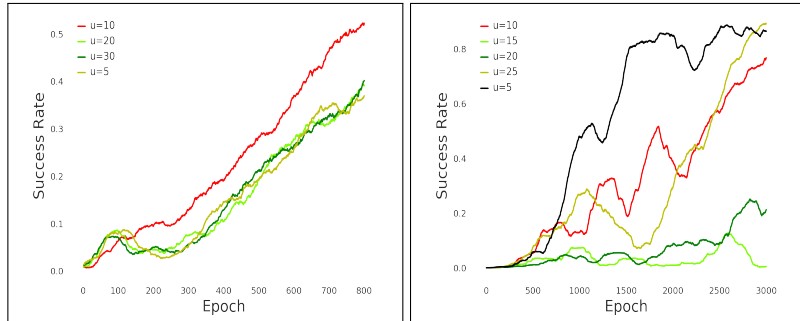

Figure 5: Column 1(maze navigation environment) and column 2(pick and place environment) show success rate performance plots of CRISP for different values of hyperparameter $u$, plotted against number of training epochs.

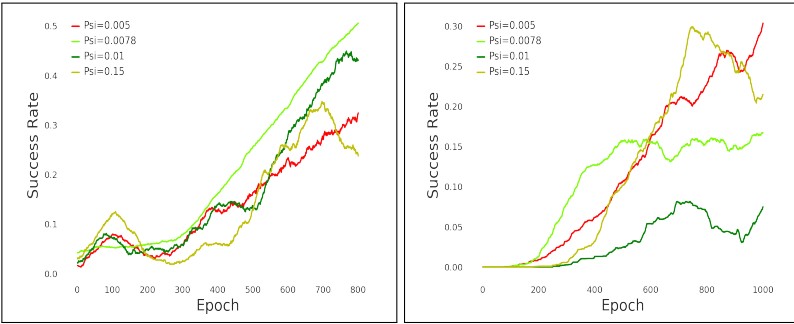

Figure 6: The success rate plots show the performance of CRISP for various values of imitation learning weight parameter $\psi$ versus number of training epochs. Column 1 shows experiments on maze navigation environment and column 2 shows experiments on pick and place environment.

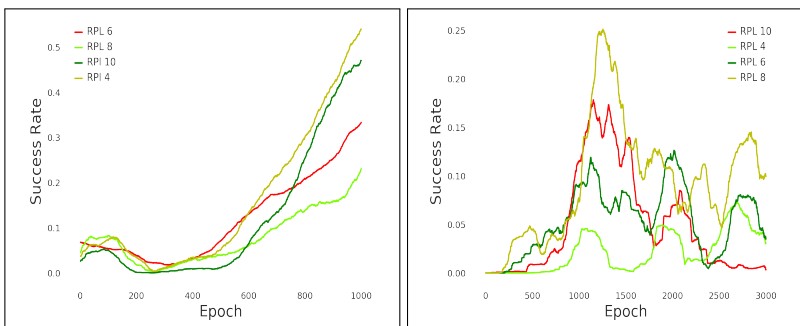

Figure 7: The success rate plots show the performance of RPL for various values of $k$ window size parameter versus number of training epochs. Column 1 shows experiments on maze navigation environment and column 2 should experiments on pick and place environment.

### A.4 GENERATING EXPERT DEMONSTRATIONS FOR MAZE NAVIGATION TASK

We use the path planning RRT Lavalle (1998) algorithm to generate optimal paths $P = (p_t, p_{t+1}, p_{t+2}, ...p_n)$ from the current state to the goal state. RRT has privileged information about the obstacle position which is provided to the methods through state consisting sparse maze array. Using these expert paths, we generate state-action expert demonstration dataset for the lower level policy.

### A.5 GENERATING EXPERT DEMONSTRATIONS FOR PICK AND PLACE TASK

In order to generate expert demonstrations, we used a human expert to perform the pick and place task in virtual reality based Mujoco simulation. In this task, an expert first picks up the block using robotic gripper, and then takes it to the target goal position. Using these expert trajectories, we generate state-action expert demonstration dataset for the lower level policy.

### A.6 ROBOTIC ROPE MANIPULATION ENVIRONMENT

Here we provide details about the rope manipulation environment, its implementation and results.

### A.6.1 ENVIRONMENT SETUP

In the robotic rope manipulation task, a deformable rope is kept on the table and the robotic arm performs pokes to nudge the rope towards the desired goal rope configuration. The task horizon is fixed at 25 pokes. The deformable rope is formed from 15 constituent cylinders, held together by joints.

Table 2: Success rates: Rope manipulation environment

| Method | Rope Manipulation |
|---|---|
| CRISP-IRL | **0.29 ± 0.04** |
| CRISP-BC | **0.32 ± 0.056** |
| RPL | 0.18 ± 0.12 |
| Hier | 0.1 ± 0.07 |

### A.6.2 PRETRAINING TASK FOR THE LOWER LEVEL PRIMITIVE

In this complex environment, we first pretrain the lower level primitive using simpler goal rope configurations which can be achieved within a few pokes (the simple goal rope configurations are chosen based on L2 distance between the initial and goal rope configurations). We specially pretrained the lower primitive on a simpler task in rope manipulation environment since without pretraining, the methods failed to provide significant results. In order to ascertain fair comparisons, we kept this pre-training requirement consistent among all the baselines in this environment. After pre-training the lower primitive, the lower primitive is used with our method CRISP.

### A.6.3 GENERATING EXPERT DEMONSTRATIONS

In complex environments, we generally do not have access to lower level expert demonstrations. Moreover, hard coding an expert policy may generate sub-optimal expert demonstrations. In rope environment we did not have access to lower level expert demonstrations. However, recall that our method requires only expert state demonstrations and not expert action demonstrations. For generating expert state demonstrations, we used an interpolation based approach, where we obtain subgoals $s_{g_i}$ by linearly interpolating between the starting rope configuration $s$ and the final rope configuration $g$ as:

$$s_{g_i} = \frac{i}{N}g + \frac{N-i}{N}s, \quad i \in \{1, 2, \ldots, N-1\} \tag{20}$$

We found $N = 15$ to perform well empirically. After generating these interpolations, we performed simple transformations to assure that the interpolations are valid rope configurations. Note that since we do not have access to lower level expert action demonstrations, we do not use expert actions demonstrations dataset to train the lower level in CRISP and other baselines.

### A.6.4 IMPLEMENTATION DETAILS

The following implementation details refer to both the higher and lower level polices, unless otherwise explicitly stated. The state and action spaces in the environment are continuous. The actor, critic and discriminator networks are formulated as 3 layer fully connected neural networks with 512 neurons in each layer. The state space for the rope manipulation environment is a vector formed by concatenation of the intermediate joint positions. The upper level predicts subgoal $s_g$ for the lower primitive. The action space of the poke is $(x, y, \eta)$, where $(x, y)$ is the initial position of the poke, and $\eta$ is the angle describing the direction of the poke. We fix the poke length to be $0.08$. While training our hierarchical approach, we select 100 randomly generated initial and final rope configurations each for training, testing and validation. For selecting train, test and validation configurations, we first randomly generate 300 distinct configurations, and then randomly divide them into 100 train, test and validation mazes each. Each experiment is run on 4 parallel workers. We use off-policy Soft Actor Critic (Haarnoja et al., 2018b) algorithm for optimizing RL objective in our experiments. We keep the regularization weight hyperparameter as $\lambda = 0.01$ in our experiments.

### A.6.5 RESULTS

In Figure 8, we show the subgoals predicted by our method in at various timesteps, thereby generating a curriculum of subgoals for the higher level policy. In Table 2, we report the success rate performance of the proposed methods and other baselines, averaged over 3 seeds. CRISP-IRL is CRISP with Inverse RL(IRL) regularization objective, and CRISP-BC is CRISP with behavior cloning(BC) regularization. While training and testing, we evaluate success rates over $N = 100$ random episodic rollouts. Since the lower level expert action demonstrations were not available,

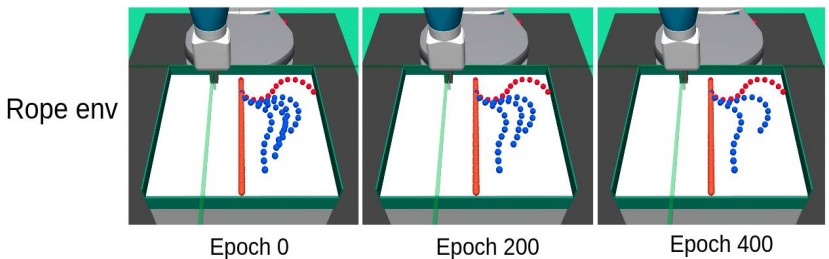

Figure 8: This figure depicts CRISP subgoals progress vs training epochs elapsed in rope manipulation environment. The higher level policy generates a curriculum of subgoals for lower primitive(walls:yellow, final goal:red, subgoals:blue)

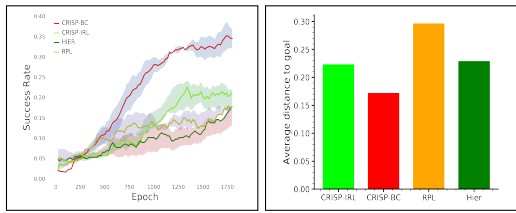

Figure 9: The success rate plots show performance comparison between our method and baselines in rope manipulation environment (Column 1) versus number of training epochs. Columns 2 plot compare the methods via distance metric(average distance between achieved goal and final goal configuration in 100 episodic rollouts). As can be seen, our methods show impressive performance compared to baselines.

we do not compute the DAC baseline. From Table 2 it is apparent that CRISP-BC and CRISP-IRL clearly outperform the baselines by a large margin. This provides convincing evidence that stable hierarchical learning indeed demonstrate better performance on complex long horizon tasks. The success rate comparison plots are provided in Figure 9

