# OpenReview forum: "CRISP: Curriculum inducing Primitive Informed Subgoal Prediction for Hierarchical Reinforcement Learning"
_ICLR.cc/2023/Conference — Submitted to ICLR 2023_

### Official Review · Reviewer_oc7V · 2022-10-24

**Confidence:** 4
**Correctness:** 3
**Technical Novelty And Significance:** 3
**Empirical Novelty And Significance:** 2
**Recommendation:** 6

**Clarity, Quality, Novelty And Reproducibility:**

The work is high quality, providing a somewhat novel hindsight-based hierarchical algorithm in the context of supervised learning, and deriving some limited analysis on the divergence between the learned policy and the expert policy. It also provides insight in a sizable number of add-ons and hacks that allow this method to work in simulated robotics tasks, which can be useful for HRL researchers trying to use similar methods. It is formatted clearly and the main ideas are expressed well, although as is typical for HRL, where the benefit comes from can be somewhat obfuscated.

**Strength And Weaknesses:**

Strengths:

The paper is relatively easy to understand and the concept is an interesting application of hindsight in learning from demonstration.

THis paper represents a fairly sizable combination of techniques to produce results.

Weaknesses:

This method is limited by the requirement of being able to reset the environment to a state mid-trajectory, which either requires a carefully designed environment or access to a simulator. This can impact the application of this method to real-world tasks.

The paper combines ideas from a wide range of areas which obfuscates where the improvements come from or if a single core idea is in particular more compelling. In particular, the primitive informed parsing could be performed without an expert and have a similar effect of curriculum, still requiring the agent to move to this information. Including the expert and thus IRL/behavior cloning, this makes it less clear why these components are inherently necessary to the core idea (or resetting hindsight labels). In addition, adding on a hierarchical meta-policy seems to only add an extra layer of complexity.

In the movement domains, there is only marginal improvement, and these domains are simple enough to be toy domains. The majority of the improvements are in the pick and place domain, where it appears that both the hierarchical and RPL methods completely fail. In this case it seems like this is because pick-and-place tasks require expert information, and a basic application of that data to policy learning would probably result in a significant improvement in performance since pick and place tasks have certainly been performed by hierarchical, RL and IRL methods in the past.

**Summary Of The Paper:**

Learn temporally abstracted skills from expert demonstrations. This work uses a two-level hierarchy, where the subgoals given to the lower level of the hierarchy are always achievable, and progressively increase in difficulty. This is achieved by choosing subgoals of increasing length for the agent until the agent is unable to achieve the subgoal.

**Summary Of The Review:**

I propose to accept this paper because it has sufficient novelty, is well written and provides interesting insights into getting HRL to work in some simulated robotics domains.

---

> ### Author Response · Authors · 2022-11-11
> **Author Response**
>
> We thank the reviewer for their detailed constructive feedback. We address the reviewer’s concerns as follows:
>
> **"The paper combines ideas from a wide range of areas which obfuscates where the improvements come from or if a single core idea is in particular more compelling."**
>
> RESPONSE:
>
> We agree that we combine many ideas to generate a general purpose method. However, in order to demonstrate the source of performance improvements, we perform baseline comparisons in Section 5.4, as shown in Figure 3. Using these baselines, we show that our method outperforms:
>
> 1. single level approaches that use expert demonstrations(DAC),
> 2. multi level hierarchical approaches(Hier), and
> 3. multi level hierarchical approaches using fixed window based parsing to relabel expert demonstrations(RPL).
>
> We hope that after considering these experiments, it becomes more clear why our method outperforms various baselines, and why various design choices are relevant.
>
> **"In particular, the primitive informed parsing could be performed without an expert and have a similar effect of curriculum, still requiring the agent to move to this information."**
>
> RESPONSE:
>
> We would like to respectfully point out that although PIP could be performed on a randomly generated trajectory generated without an expert, the reachable subgoals thus generated will not be "good" subgoals which motivate the higher level to predict efficient subgoals for achieving the final goal. In the worst case, the higher level policy might predict degenerate subgoals to the lower primitive which are always achievable, but provide no significant improvement towards achieving the final goal.
> Update: We have performed an additional ablation study where we evaluate the effect of adding sub-optimal or “bad” demonstrations to the expert dataset and evaluate its effect on learning and performance. In the experiment, we incrementally increased the number of “bad” demonstrations (e.g. 10, 30, 50, 70 etc) and plotted the success rates of our CRISP-IRL against the number of epochs. Here, for generating “bad” demonstrations, we randomly picked a trajectory and ran our primitive parsing method PIP for selecting subgoals. This intuitively meant that PIP would not be able to select "good" subgoals for generating higher level expert subgoal dataset. We empirically verified that the performance degrades as we increment the number of bad demonstrations in the expert demonstration dataset. We promise to add this experiment in the final draft. This empirically demonstrates that we need expert demonstrations to guide the higher level policy to predict "good" subgoals.
>
>
> **"In the movement domains, there is only marginal improvement, and these domains are simple enough to be toy domains."**
>
> RESPONSE:
>
> In the maze navigation domain, we use sparse rewards to make the environment more challenging. Additionally, we randomize the walls and gates in the four room environments, which forces the agent to consider the maze while predicting actions on the environment(explained in detail in Section 5.2.1). This significantly increases the environment complexity and the methods take a large number of timesteps (2.93e6) for learning to perform the task which is verified empirically.
>
> We have also performed further experimentation in a complex rope manipulation environment(please refer to Appendix A.6), and found that our method indeed outperforms the baselines in achieving complex tasks that require long term planning. We hope that these evaluations are enough to demonstrate the efficacy of our method on complex long horizon tasks.
>
> **"The majority of the improvements are in the pick and place domain, where it appears that both the hierarchical and RPL methods completely fail. In this case it seems like this is because pick-and-place tasks require expert information, and a basic application of that data to policy learning would probably result in a significant improvement in performance since pick and place tasks have certainly been performed by hierarchical, RL and IRL methods in the past."**
>
> RESPONSE:
>
> We would like to respectfully point out that the baselines RPL and DAC both use expert demonstrations. Since our method outperforms RPL, we demonstrate that adaptive parsing of expert demonstrations (used to generate higher level demonstration dataset) induced by our hierarchical curriculum learning approach outperforms fixed window parsing based approach. Since our method outperforms DAC, we demonstrate the significance of using our hierarchical formulation, as compared to the single level DAC approach.
>
> We hope that this response addresses the reviewer’s concern. Please let us know, and we will be happy to address additional concerns if any.

---

> > ### Author Response · Authors · 2022-12-08
> > **Discussion:**
> >
> > Dear reviewer,
> >
> > We hope our response clarified your initial concerns/questions. We would be happy to provide further clarifications where necessary.

---

### Official Review · Reviewer_xPx6 · 2022-10-26

**Confidence:** 4
**Correctness:** 3
**Technical Novelty And Significance:** 4
**Empirical Novelty And Significance:** 2
**Recommendation:** 5

**Clarity, Quality, Novelty And Reproducibility:**

Overall, this paper presents an original idea for an important problem in hierarchical reinforcement learning: learning effective lower-level skills. The idea is high-quality, and the inclusion of an analysis of the suboptimality of the learned skills helps to improve its quality. However, the paper is not experimentally thorough, and the quality of the paper is brought down by the overall simplicity of the tasks used for evaluation, see Weakness (1). The paper’s experimental limitations outweigh the high-quality idea, and the included suboptimality analysis.

The paper is generally clear, except for a minor issue detailed in Weakness (4) regarding how performance is reported, and in Weakness (2) regarding the data requirements of PIP. After these are resolved, I would consider the paper generally clear, with no major clarity flaws. Experiments appear to be algorithmically reproducible, as hyperparameters of the proposed method (including the introduced \lambda and u and specified), though some details from the environment setup are missing. For example, how are the doors of the maze randomized, and how is the train-test split of the evaluations tasks constructed for each task.


**Strength And Weaknesses:**

Strengths Of The Paper:

1) One way the proposed approach is appealing is due to the elimination of one form of expert annotation: using the lower-level policy to adaptively parse expert state-demonstrations into skills, rather than a fixed horizon-based parse.

2) Suboptimality bounds for the higher-level policy and lower-level policy are given, which is implied to become tighter as the lower-level skills improve as training progresses.

3) Implementation details for the algorithm on both tasks is discussed at length, and specific hyperparameter values are given which will help to facilitate reproducibility.

Weaknesses Of The Paper:

1) Experimental evaluation of the approach, while showing the proposed method in a positive light, is not thorough, and can be improved in several ways. Firstly, the method is only evaluated on two reinforcement learning tasks, both of which are based on the same position-based robotic manipulator. This environment functions similarly to a 3D pointmass, where actions are directly added to the current 3D position of the gripper. The environment is made more challenging by using a sparse reward, but the complexity of the underlying task is simple in comparison to other tasks that were studied in previous HRL papers, including locomotion-based maze navigation (from HIRO, Nachum et al. 2018), and torque-based manipulation (from HAC, Levy et al, 2018).

2) While the proposed adaptive parsing method is effective in practice, some details are unclear, which I will discuss next. First, the paper states that on the maze task the replay buffer is flushed and re-populated every 200 steps, and 100 steps for the block task. How many new environment steps are taken or required during this flushing and re-population operation, and are these included in the budget reported (2.93e6 timesteps for the maze task and 6.75e6 timesteps for the block task). For a fair comparison, baselines would ideally be compared to the proposed method under the same conditions (ie, the same number of environment timesteps, including those used by PIP). It remains possible the evaluation already takes this detail into account, but is unclear from the text.

3) On Page 6 of the manuscript, the authors state “our method deals with non-stationarity by regularizing the higher policy with imitation learning” and this is not validated. I agree with the author’s that it is likely the proposed approach does mitigate non-stationarity by design, but this claim is not experimentally verified. Such an experimental verification should be performed that compares to other HRL methods that mitigate non-stationarity. Building on Weakness (2), the authors should ensure all methods operate in the same conditions in regards to any additional samples that PIP requires to relabel the buffer.

4) Performance is reported in a somewhat uncommon format. It would be helpful to report success rate versus the cumulative number of environment samples collected (including any additional samples that may be required during the PIP parsing step, if any).


**Summary Of The Paper:**

This paper investigates the problem setting of off-policy hierarchical reinforcement learning, focusing on automatic-curriculum generation (via the proposed PIP) and bootstrapping from  a handful of expert demonstrations using imitation-based regularization. The paper’s contributions include an adaptive parsing method that segments state-only expert demonstrations into skills that are at the fringe of the current capabilities of the lower-level policy. The proposed adaptive method is shown to outperform its fixed counterpart on two simulated tasks involving a robotic manipulator moving through a maze, and moving a block to a desired goal position. Furthermore, visualizations in the paper confirm that subgoals of increasing difficulties are set by the higher-level policy as training progresses (see Figure 2, Page 3 of manuscript).


**Summary Of The Review:**

Overall, the paper presents an original and promising idea, but falls short in terms of its experimental evaluation of the idea. In addition, certain flaws related to clarity hinder the interpretability of the results (see ‘Strength And Weaknesses’). I am willing to reconsider my evaluation if weaknesses are addressed, either with new results, or clarifications to questions.

---

> ### Author Response · Authors · 2022-11-11
> **Author Response Part 1/2**
>
> We thank the reviewer for their detailed constructive feedback. We address the reviewer’s concerns as follows:
>
> **"the method is only evaluated on two reinforcement learning tasks, both of which are based on the same position-based robotic manipulator. This environment functions similarly to a 3D pointmass, where actions are directly added to the current 3D position of the gripper, "the complexity of the underlying task is simple in comparison to other tasks that were studied in previous HRL papers, including locomotion-based maze navigation (from HIRO, Nachum et al. 2018), and torque-based manipulation (from HAC, Levy et al, 2018)."**
>
> RESPONSE:
>
> We would like to point out that in locomotion-based maze navigation (from HIRO, Nachum et al. 2018), and torque-based manipulation (from HAC, Levy et al, 2018) environments, it is hard to generate expert demonstrations. Since our method performs data relabeling on expert demonstrations to generate subgoal dataset, we require expert state demonstrations. In order to show that our method works in complex environments, we perform experiments on our novel Rope Manipulation environment, which we have added in detail in Appendix A.6. It is hard to generate expert state-action demonstrations in this environment, and a hard coded expert policy might generate sub-optimal expert trajectories. But since our method only requires expert state demonstrations and not expert action demonstrations, we use an interpolation based method to generate expert state rope configurations and pass it to PIP. We perform performance comparisons in Table 2 and Figure 8 and show that CRISP clearly outperforms other baselines. Please refer to Appendix A.6 for more details. We would also like to point out that methods like HIRO and HAC can be used in conjunction with CRISP. We propose to first use methods like HIRO and HAC to generate expert demonstration trajectories, and then use our method CRISP to leverage the generated expert dataset to solve complex long horizon task using hierarchical curriculum learning. This is an interesting idea that we would like to explore further in future work.
>
>
> **"How many new environment steps are taken or required during this flushing and re-population operation, and are these included in the budget reported (2.93e6 timesteps for the maze task and 6.75e6 timesteps for the block task)."**
>
> RESPONSE:
>
> Let us consider the task horizon to be of length $L$ and the number of expert demonstrations to be $N_D$. The environment steps during re-population operation are thus $L*N_D$. It is to be noted that the environment steps taken during the repopulation operation are a factor of the number of expert demonstrations that are relabeled (i.e $N_D$). These steps are indeed included in the budget reported (2.93e6 timesteps for the maze task and 6.75e6 timesteps for the block task).
>
> **"For a fair comparison, baselines would ideally be compared to the proposed method under the same conditions (ie, the same number of environment timesteps, including those used by PIP). It remains possible the evaluation already takes this detail into account, but is unclear from the text.",** and
>
> **"Performance is reported in a somewhat uncommon format. It would be helpful to report success rate versus the cumulative number of environment samples collected (including any additional samples that may be required during the PIP parsing step, if any)."**
>
> RESPONSE:
>
> Our method CRISP re-populates the demonstration dataset by relabeling expert demonstrations using the current lower level primitive. We agree that while re-populating the buffer, we perform extra environment steps($L*N_D$), which can be considered a limitation of our approach. We would like to point out that if the baselines are allowed to train while CRISP re-populates the dataset, then the success rate plot comparisons would not be fair. To ascertain fair comparison, we choose to plot the success rates against the number of training epochs, instead of "cumulative number of environment samples collected". Thus, although CRISP requires extra environment steps for re-labeling the expert dataset, the success rates are plotted consistently after each training epoch for all the methods. We hope that this clearly explains our motivation for the current choice of success rate plots.

---

> > ### Author Response · Authors · 2022-11-11
> > **Author Response Part 2/2**
> >
> >
> > **"On Page 6 of the manuscript, the authors state “our method deals with non-stationarity by regularizing the higher policy with imitation learning” and this is not validated. I agree with the author’s that it is likely the proposed approach does mitigate non-stationarity by design, but this claim is not experimentally verified. Such an experimental verification should be performed that compares to other HRL methods that mitigate non-stationarity. Building on Weakness (2), the authors should ensure all methods operate in the same conditions in regards to any additional samples that PIP requires to relabel the buffer."**
> >
> > RESPONSE:
> >
> > We agree to the issue pointed out by the reviewer that although our method intuitively mitigates non-stationarity, we do not empirically compare with other methods which deal with non-stationarity, e.g. HIRO(Nachum et al. 2018) and HAC(Levy et al. 2018). We want to point out that while these methods perform data relabeling, they do not leverage expert demonstrations to solve long horizon tasks. In order to guarantee a fair comparison, we will have to re-implement these methods with additional imitation learning objectives. While this is an interesting avenue for future work, we have not performed comparisons with these methods in the current work.
> >
> > **"though some details from the environment setup are missing. For example, how are the doors of the maze randomized, and how is the train-test split of the evaluations tasks constructed for each task."**
> >
> > RESPONSE:
> >
> > We address the reviewer's concern by providing the above details in Section 5.2.2 and Section 5.3.2. We also describe the details here:
> >
> > "how are the doors of the maze randomized":
> >
> > The table is discretized into a rectangular $W*H$ grid, and the vertical and horizontal wall positions $W_{P}$ and $H_{P}$ are randomly picked from $(1,W-2)$ and $(1,H-2)$ respectively. In the four room environment thus constructed, the four gate positions are randomly picked from $(1,W_{P}-1)$, $(W_{P}+1,W-2)$, $(1,H_{P}-1)$ and $(H_{P}+1,H-2)$.
> >
> > "how is the train-test split of the evaluations tasks constructed for each task":
> >
> > Maze navigation environment:
> >
> > We select $100$ randomly generated mazes each for training, testing and validation. For selecting train, test and validation mazes, we first randomly generate $300$ distinct mazes, and then randomly divide them into $100$ train, test and validation mazes each.
> >
> > Pick and place environment:
> >
> > While training, we select $100$ randomly pick and place environments each for training, testing and validation. For selecting train, test and validation mazes, we first randomly generate $300$ distinct environments with different block and target goal positions, and then randomly divide them into $100$ train, test and validation mazes each.
> >
> > We hope that this response addresses the reviewer’s concern. Please let us know, and we will be happy to address additional concerns if any.

---

> > > ### Author Response · Authors · 2022-12-08
> > > **Discussion:**
> > >
> > > Dear reviewer,
> > >
> > > We hope our response clarified your initial concerns/questions. We would be happy to provide further clarifications where necessary.

---

### Official Review · Reviewer_WG9D · 2022-10-27

**Confidence:** 3
**Correctness:** 3
**Technical Novelty And Significance:** 3
**Empirical Novelty And Significance:** 2
**Recommendation:** 5

**Clarity, Quality, Novelty And Reproducibility:**

While motivation and related work are written clearly, the method section needs clarifications and improvements. The experiments are mostly clear, but some information (e.g., on the expert trajectories) are missing to fully understand the experiments and be able to reproduce them. There are too many design details (optimizer, learning rate, …) missing, to guarantee reproducibility. However, this could be fixed by providing code to run the experiments. The idea seems novel.

**Strength And Weaknesses:**

Strengths:
+ The relabelling of the expert demonstration states based on failures to reach them is a very interesting idea.  The emergence of a learning curriculum from this simple procedure seems very promising and is nicely visualized in Figure 2.
+ The results compared to the baselines on the two investigated tasks look impressive and highlight the strength of using a hierarchy combined with the subgoal learning curriculum.

Weaknesses:
- The method part is at different parts very hard to follow, and I found various parts confusing. Below is a list of points that I think need to be addressed:
    • p. 3, section 3.1: The lower primitive policy $\pi^j_L$ is introduced with superscript $j$, where $j$ denotes "an indicator of the current subgoal reaching capability of the lower primitive". In my opinion this sentence does not sufficiently explain $j$. The indicator $j$ is not further clarified, does not change in the pseudocode, and is dropped later (section 3.2). So why is the superscript $j$ needed?
    • p. 4, Definition 1: What is $D_{TV}$? I'm assuming total variation, but it is not introduced.
    • p. 4, The proof for the suboptimality of the lower primitive is missing. It can probably be derived from the proof for the upper policy, however, seeing that they differ with respect to lambda it is necessary to explain where this difference comes from.
    • p. 5, Eq. 10: What is $\epsilon$? This needs to be introduced.
    • p. 5, section 3.4: Why does Eq. 10 incentivize the higher policy to make "reasonable progress towards achieving the final goal"? Does it not just incentivize generating subgoals that are hard to distinguish from the subgoals of dataset $D_g$?
    • p. 5, section 3.5: It reads like the off-policy RL objectives can be referred to as $J^H_D$ and $J^L_D$. Shouldn't the off-policy objective be $J^H_{\theta_H}$ and J^L_{\theta_L} of Eq. 11 and Eq. 12? Because $J^H_D$ is defined above as the IRL objective (Eq. 10). $J_D^L$ is not defined before. The authors need to be more precise about the different objectives.
    • p. 5, Eq. 11 & 12: $\lambda$ is here introduced as a hyperparameter whereas in section 3.2, $\lambda$ was used as a factor for the upper bound of suboptimality. To avoid confusion different names should be used.
- From the experiments it is not clear how much the approach relies on the expert trajectories to learn to solve the tasks. How many expert trajectories were used for each of the two tasks? How were the expert trajectories generated? How does the number of expert demonstrations influence the system's performance? For example, how do fewer or more demonstrations affect the performance? Could the system deal with "bad expert demonstrations" that are not useful in solving the task? For example, what would happen when a couple of random rollouts are included in the demonstration dataset? I think these questions need clarification in the description of the experiments, further discussion, and maybe additional experiments.
- What is the effect of resetting the subgoal buffer, i.e., the hyperparameter $u$? How does $u$ affect the sample efficiency of the approach? An ablative study evaluating different values of $u$ would help understand how resetting the subgoal dataset affects the method.
- How is CRISP-BC different to CRISP-IRL? Why does it perform better than CRISP-IRL in the Pick and Place task? Highlighting the difference between the two versions would also help understanding the approach.
- The method requires that the simulator can be reset to an arbitrary state. This is a major restriction for applying the method. The authors acknowledge this limitation but a more detailed discussion on how this limitation could be weakened would improve the paper. For example, it prohibits applying the system on real robots. Thus, the claim of the abstract that the method “[…] is suitable for most real world robotic control tasks” is too strong.
- Please fix the citation style. Use citet only of the citation is part sentence, grammatically (as a subject or object). Otherwise ,use \citep. Here an example: “Learning effective hierarchies of policies has garnered substantial research interest in RL Barto & Mahadevan (2003)” should be “Learning effective hierarchies of policies has garnered substantial research interest in RL (Barto & Mahadevan, 2003)”.
Minor corrections and suggestions:
- typo on p. 1: "balnced"
- bottom of p. 5: it should be Algorithm 2 instead of Algorithm 0
- typo on p. 9: "tn maze navigation"
- Figure 1 should be referenced in the text.
- introduce the abbreviation IRL on p. 3 when first mentioning inverse reinforcement learning

**Summary Of The Paper:**

The manuscript presents CRISP, an approach for hierarchical reinforcement learning from expert trajectories. Their approach relabels states from expert demonstration as subgoals to train a higher level. States are labelled as subgoals when they are the first in a subsequence that can’t be reached by the lower level from the start of this subsequence. Thus, subgoals are chosen that are “just out of reach” for the lower level, developing a learning curriculum. The two levels are trained to optimize both offline reinforcement learning and imitation learning within one objective. The paper provides a bound for the suboptimality of the method compared to an optimal policy. The method is evaluated on two simulated tasks and outperforms two hierarchical reinforcement learning baselines.

**Summary Of The Review:**

In sum, I think the idea is very interesting, but the clarity and details of the method and experiments still need improvement to be accepted. The idea seems interesting, but it’s potential and limitations could be evaluated much more thoroughly in experiments.

-- post rebuttal update --
See the answers below. I have raised my score to 5 (and updated correctness to 3).

---

> ### Author Response · Authors · 2022-11-11
> **Author Response Part 1/2**
>
> We thank the reviewer for their detailed constructive feedback. We address the reviewer’s concerns as follows:
>
> **"p. 3, section 3.1: The lower primitive policy $\pi_{L}^{j}$ is introduced with superscript $j$, where $j$ denotes "an indicator of the current subgoal reaching capability of the lower primitive". In my opinion this sentence does not sufficiently explain $j$. The indicator is not further clarified, does not change in the pseudocode, and is dropped later section 3.2). So why is the superscript $j$ needed?"**
>
> RESPONSE:
>
> We agree that although we meant to use superscript $j$ to intuitively explain the current goal reaching capability of lower primitive, it might add to confusion since it is not of empirical significance. Therefore, we drop the superscript in the paper draft.
>
> **"p. 4, Definition 1: What is $D_{TV}$? I'm assuming total variation, but it is not introduced."**
>
> RESPONSE:
>
> $D_{TV}$ is indeed TV divergence. We have added its introduction in the paper draft.
>
> **"p. 4, The proof for the suboptimality of the lower primitive is missing. It can probably be derived from the proof for the upper policy, however, seeing that they differ with respect to lambda it is necessary to explain where this difference comes from.”**
>
> RESPONSE:
>
> We have now added the proof of both the higher level and lower level policy in Appendix A.1 and A.2 respectively, clearly elucidating the differences between the two proofs.
>
> **"p. 5, Eq. 10: What is $\epsilon$? This needs to be introduced"**
>
> RESPONSE:
>
> $\epsilon$ denotes the parameters of the higher level discriminator. We have added its definition in the paper draft.
>
> **"p. 5, section 3.4: Why does Eq. 10 incentivize the higher policy to make "reasonable progress towards achieving the final goal"? Does it not just incentivize generating subgoals that are hard to distinguish from the subgoals of dataset $D_g$?"**
>
> RESPONSE:
>
> (Please note that as we have moved the proofs to appendix, the previous Eq 10 in the reviewer’s concern has become Eq 5. We apologize for the confusion caused if any.)
>
> We agree that this statement might lead to confusion, and have therefore changed this statement from the draft. To clarify, Eq 5 incentivizes **objective B**(generating subgoals that are hard to distinguish from the subgoals of dataset $D_g$), and Eq 6 jointly incentivizes the higher policy for **objective A**(generating subgoals that achieve the final goal) and **objective B**.
>
> **"p. 5, section 3.5: It reads like the off-policy RL objectives can be referred to as $J_{D}^{H}$ and $J_{D}^{L}$. Shouldn't the off-policy objective be $J_{\theta_H}^{H}$ and $J^L_{\theta_L}$ of Eq. 11 and Eq. 12? Because $J_{D}^{H}$ is defined above as the IRL objective (Eq. 10). $J_{D}^{L}$ is not defined before. The authors need to be more precise about the different objectives.”**
>
> RESPONSE:
>
> We thank the reviewer for pointing out this error. We have corrected this in the paper draft. The off-policy RL objectives are indeed $J_{\theta_H}^{H}$ and $J^L_{\theta_L}$.
>
> **"p. 5, Eq. 11 & 12: $\lambda$ is here introduced as a hyperparameter whereas in section 3.2, $\lambda$ was used as a factor for the upper bound of suboptimality. To avoid confusion different names should be used."**
>
> RESPONSE:
>
> To avoid any confusion, we have renamed the hyperparameter $\lambda$ with a different symbol $\Psi$.
>
> **"From the experiments it is not clear how much the approach relies on the expert trajectories to learn to solve the tasks. How many expert trajectories were used for each of the two tasks? How were the expert trajectories generated? How does the number of expert demonstrations influence the system's performance? For example, how do fewer or more demonstrations affect the performance?"**
>
> RESPONSE:
>
> To address the reviewer's concern, we have added the discussion and ablation experiments on using expert demonstrations in Section 5.4 and Figure 4. We also describe the rationale here:
> The number of expert demonstrations for each task is kept to be $100$ after performing ablation experiments as provided in Figure 4 in the appendix. If the number of demonstrations is less, the policy might overfit to the demonstrations, which hampers the overall performance. This is clearly depicted in Figure 4, which shows that the performance suffers if we employ a small number of expert demonstrations. Although the number of available expert demonstrations is generally dependent on the intended task environment, in our experiments, we increased the expert demonstrations until there was no significant improvement.

---

> > ### Author Response · Authors · 2022-11-11
> > **Author Response Part 2/2**
> >
> > **"Could the system deal with "bad expert demonstrations" that are not useful in solving the task? For example, what would happen when a couple of random rollouts are included in the demonstration dataset? I think these questions need clarification in the description of the experiments, further discussion, and maybe additional experiments."**
> >
> > RESPONSE:
> >
> > It is important to note that since CRISP uses PIP to select good subgoals from lower level expert dataset, we require "good" lower level expert demonstration trajectories. If the expert trajectory is "bad", PIP is unable to select good subgoals for higher level policy, leading to poor performance.
> >
> > **"What is the effect of resetting the subgoal buffer, i.e., the hyperparameter $u$? How does $u$ affect the sample efficiency of the approach? An ablative study evaluating different values of would help understand how resetting the subgoal dataset affects the method."**
> >
> > RESPONSE:
> >
> > To address the reviewer's concern, we have added an ablative study for varying the value of $u$ hyperparameter in Section 5.4 and Figure 5.
> >
> > **"How is CRISP-BC different to CRISP-IRL? Why does it perform better than CRISP-IRL in the Pick and Place task? Highlighting the difference between the two versions would also help understanding the approach."**
> >
> > RESPONSE:
> >
> > CRISP-IRL denotes CRISP with inverse reinforcement learning(IRL) regularizer, whereas CRISP-BC denotes CRISP with behavior cloning(BC) regularizer. We have added the definitions in the paper draft as well.
> >
> > Update: After further experiments and carefully selecting hyperparameter $u$ for the pick and place task, we found that CRISP-IRL outperforms CRISP-BC. We have updated the results in Table 1 and Figure 3 column 2.
> >
> > Rationale: Our motivation for using IRL was that although behavior cloning works efficiently in many scenarios, it fails to perform in problems that require long term planning. We can see clearly from Table 1 that this indeed is the case in maze navigation and pick and place environments, where CRISP-IRL performs better than CRISP-BC. Additionally, we would like to add that in our experiments, we found that the advantages of inverse reinforcement learning are sometimes counterbalanced by the fact that they are really difficult to train. We were able to achieve good results in maze navigation and pick and place domains, but we did further experimentation in a novel rope environment(Please refer to Appendix A.6) and found that BC performs slightly better than IRL:
> >
> >
> > Success Rates in rope navigation:
> >
> > **CRISP-IRL: 0.29 $\pm$ 0.04**
> >
> > **CRISP-BC: 0.32 $\pm$ 0.056**
> >
> > We therefore would like to add that although in general CRISP-IRL works better than CRISP-BC in long horizon complex tasks, CRISP-BC works better in cases when it is hard to train an IRL objective.
> >
> > **"The method requires that the simulator can be reset to an arbitrary state. This is a major restriction for applying the method. The authors acknowledge this limitation but a more detailed discussion on how this limitation could be weakened would improve the paper. For example, it prohibits applying the system on real robots. Thus, the claim of the abstract that the method “[…] is suitable for most real world robotic control tasks” is too strong."**
> >
> > RESPONSE:
> >
> > We provide discussion on possible methods to relax this assumption in Section 6. As suggested by the reviewer, we agree and drop the claim “is suitable for most real world robotic control tasks” in the paper draft.
> >
> > As suggested by the reviewer as "Minor suggestions and corrections", we have fixed the citation style and the other typos in the paper draft.
> >
> > **"There are too many design details (optimizer, learning rate, …) missing, to guarantee reproducibility. However, this could be fixed by providing code to run the experiments."**
> >
> > RESPONSE:
> >
> > To address the reviewer’s concerns, we have updated the implementation details in Section 5.2.2 and Section 5.3.2 to extensively encompass the environment and hyperparameter details. We also plan to release open source implementation of our method and experiments with relevant implementation details.
> >
> > We hope that this response addresses the reviewer’s concern. Please let us know, and we will be happy to address additional concerns if any.

---

> > > ### Comment · Reviewer_WG9D · 2022-11-28
> > > **Answer to Authors**
> > >
> > > Thank you for the detailed answer. After reading the modified draft, the authors’ response and the other reviewers’ comments, I have raised my score.
> > >
> > > The methodology section was sufficiently improved with the corrections and clarifications. Additionally, the new implementation details improve reproducibility.
> > >
> > > The discussion here on the differences between CRISP-BC and CRISP-IRL is, to some extent, helpful. While the novel rope experiment seems interesting, it is not clear to me from the description so far, why this is a task in which it is hard to train an IRL objective. Thus, I believe the explanation of CRIPS-IRL’s better performance is not well supported. Additionally, to make the paper self-contained, the BC loss and the explanations for IRL's better performance should have been added to the paper.
> > >
> > > Besides that, the additional ablations and evaluations are highly appreciated. However, these ablations could have been more thorough and still leave some open questions to me. How much the method relies on the quality of the expert trajectories could have been quantitatively evaluated. Also, it is unclear why $u=100$ and $u=50$ were used in the experiments while the ablation studies examine values of $u < 30$. The ablation also seems to suggest that small $u$ values are a better choice for the two experiments.
> > > In sum, while the paper gained in clarity and reproducibility, to me there are still some missing pieces to completely understand the crucial design choices that make this method work.

---

> > > > ### Author Response · Authors · 2022-11-29
> > > > **Author Response**
> > > >
> > > > We thank the reviewer for their helpful responses. We address the reviewer’s concerns as follows:
> > > >
> > > > **"While the novel rope experiment seems interesting, it is not clear to me from the description so far, why this is a task in which it is hard to train an IRL objective"**
> > > >
> > > > RESPONSE:
> > > >
> > > > As explained in Appendix A.6, the rope manipulation environment is a complex environment, where the gripper performs pokes on a rope to change its current configuration to goal rope configuration. Moreover, as explained in section A.6.3, since generating expert demonstrations is hard in this complex environment, we use an interpolation based approach to generate expert state demonstrations. This may generate sub-optimal expert trajectories (which is in contrast with generating expert trajectories for maze navigation or pick and place environments, where it is easier to generate expert trajectories owing to the simplicity of the task, as mentioned in appendix A.4 and A.5).
> > > >
> > > > For these reasons, although in general CRISP-IRL works better than CRISP-BC in maze navigation task (environment: simple, generating expert demonstrations: easy) and pick and place task (environment: complex, generating expert demonstrations: easy), CRISP-IRL is hard to train in scenarios like the rope manipulation environment (environment: complex, generating expert demonstrations: hard), where the generated expert trajectories may be sub-optimal.
> > > >
> > > > We would also like to additionally point out that when comparing CRISP-IRL and CRISP-BC in rope manipulation environment, the difference in performance is quite small (ref Table 2):
> > > >
> > > >
> > > > Success Rates in rope navigation:
> > > >
> > > > **CRISP-IRL: 0.29 $\pm$ 0.04**
> > > >
> > > > **CRISP-BC: 0.32 $\pm$ 0.056**
> > > >
> > > >
> > > > Thus although CRISP-BC outperforms CRISP-IRL, the performance difference is not very large.
> > > >
> > > >
> > > > **"Additionally, to make the paper self-contained, the BC loss and the explanations for IRL's better performance should have been added to the paper."**
> > > >
> > > > RESPONSE:
> > > >
> > > > We agree that adding this discussion will further clarify the difference between CRISP-IRL and CRISP-BC approaches and provide a rationale to their relative performance. We promise to add this detailed discussion in the final draft.
> > > >
> > > > **"How much the method relies on the quality of the expert trajectories could have been quantitatively evaluated"**
> > > >
> > > > RESPONSE:
> > > >
> > > > As mentioned in a previous response, we have added the ablation experiment where we quantitatively evaluate the effect of varying the number of expert demonstrations in Section 5.4 and Figure 4. Due to time constraints, we could not evaluate the effect of adding sub-optimal or “bad” demonstrations to the expert dataset and evaluate its effect on learning and performance (although we added a discussion in Section 5.4 ablative studies). We promise to conduct this experiment and add it in the ablative studies in the final draft.
> > > >
> > > > **"it is unclear why $u=100$ and $u=50$ were used in the experiments while the ablation studies examine values of $u<30$"**
> > > >
> > > > RESPONSE:
> > > >
> > > > In the experiment section, we had reported $u=100$ and $u=50$ against the number of training iterations and in the appendix section we had reported $u=10$ and $u=5$ against the number of epochs. Thus in the ablative studies, the reported $u$ should be multiplied by a factor of $10$ for consistency. We understand now that this causes confusion and we apologize for using different metrics. We promise to make the metrics consistent throughout both the sections in the final draft.
> > > >
> > > > We hope that this response addresses the reviewer’s concern. Please let us know, and we will be happy to address additional concerns if any.

---

> > > > > ### Author Response · Authors · 2022-12-02
> > > > > **Author Response: Ablation Experiment**
> > > > >
> > > > > **Update:** We were able to conduct the experiment where we evaluate the effect of adding sub-optimal or “bad” demonstrations to the expert dataset and evaluate its effect on learning and performance.
> > > > >
> > > > > Experiment Details: Out of 100 expert lower level demonstrations in total, we incrementally increased the number of “bad” demonstrations (e.g. 10, 30, 50, 70 etc) and plotted the success rates of our CRISP-IRL against the number of epochs. Here, for generating “bad” demonstrations, we randomly picked a trajectory and ran our primitive parsing method PIP for selecting subgoals. This intuitively meant that PIP would not be able to select "good" subgoals for generating higher level expert subgoal dataset.
> > > > >
> > > > > We empirically verified that the performance degrades as we increment the number of bad demonstrations in the expert demonstration dataset. Currently we have been able to perform experiments using CRISP-IRL, and we will also perform further experiments using CRISP-BC.
> > > > >
> > > > > We promise to add the generated success rate graphs and their detailed explanation in the final draft.

---

> > > > > > ### Author Response · Authors · 2022-12-08
> > > > > > **Discussion:**
> > > > > >
> > > > > > Dear reviewer,
> > > > > >
> > > > > > We hope our response clarified your initial concerns/questions. We would be happy to provide further clarifications where necessary.

---

### Author Response · Authors · 2022-12-08
**General response to reviewers and AC, and summary of updates to manuscript:**

**Summary of updates/revisions:**

We thank the reviewers for their detailed feedback that is well organized, and easy to follow. We summarize the high level updates to the manuscript to address the reviewers' feedback and suggestions:

1. In order to demonstrate the efficacy of the proposed approach on complex long horizon tasks, we have additionally performed and added experiments on novel rope manipulation environment in the manuscript, which we have added in detail in Appendix A.6. We have also added ablation experiments in this environment in Appendix A.6.

2. We have added the following ablation experiments:

	a. we quantitatively evaluate the effect of varying the number of expert demonstrations on the performance in various environments [Section 5.4 and Figure 4].

	b. we quantitatively evaluate the effect of varying the value of hyperparameter $u$ on the performance in various environments [Section 5.4 and Figure 5].

3. We have added the sub-optimality analysis for lower level policy in appendix A.2 along with the analysis for higher level policy in appendix A.1.

4. We have improved the methodology section based on the corrections and suggestions of the reviewers.

5. As suggested by the reviewers, we have added additional details of the environment setup in all the three simulation environments. We have also added new implementation details to enhance reproducibility. We will also release open source implementation of our method and experiments with relevant implementation details.

6. We have fixed minor text issues as highlighted by the reviewers and improved the text in general by fixing minor issues.

**We briefly discuss our contributions to address common concerns of the reviewers:**

1. We propose CRISP, a general purpose lower level primitive informed method for efficient hierarchical reinforcement learning. CRISP uses our novel PIP method to perform data relabeling on expert demonstrations. We evaluate our method on three complex long horizon tasks, and demonstrate that it makes substantial gains over its baselines. Through this paper, we hope that our approach will introduce further research work in hierarchical curriculum learning based approaches.

2. We perform experiments in maze navigation, pick and place and rope manipulation environments. These environments vary both in terms of task complexity and the complexity of generating demonstrations. As shown in the experiments, CRISP consistently outperforms the baselines in all these environments, which clearly demonstrates the efficacy of our method on complex long horizon tasks.

3. We perform extensive comparisons with various baselines to demonstrate the superiority of our method over baselines. We also perform extensive ablation experiments to corroborate the significance of our design choices in Section 5.4.

---

### Decision · Program_Chairs · 2023-01-20

**Decision:**

Reject

**Justification For Why Not Higher Score:**

This paper should not receive of higher score because of the flaws in the experiments. There experiments do not properly justify better performance and there are no experiments that produce knowledge of how the new method actually works.

**Justification For Why Not Lower Score:**

N/A

**Metareview: Summary, Strengths And Weaknesses:**

This paper proposes a hierarchical approach to learning from demonstration that uses a novel data relabeling technique to periodically relabel certain states as sub-goals of the task during learning. Combined with other methods, such as a discriminator and inverse RL, the approach aims to learn a curriculum of subgoal tasks to solve the primary task. The reviewers agree that the approach shows promise and has the potential to lead to better learning from demonstration. However, there is a need for more understanding of how the method behaves and why it works well. In particular, it would be helpful to validate the components necessary for the algorithm to work effectively and if it is truly mitigating nonstationarity as claimed. Additionally, the current results are based on only three trials, making it difficult to conclude that one method is superior to another.

Overall, I encourage the authors to conduct more experiments focusing on understanding how the algorithm works rather than just its performance. For example, Figure 2 provides some insight into the curriculum learning aspects of the method, and expanding on this type of experiment for the other components and hyperparameters of the algorithm would greatly increase the reader's understanding of the method and what is necessary for creating effective hierarchical learning from demonstration systems.

**Summary Of Ac-Reviewer Meeting:**

There was no AC-reviewer meeting. I was added as an AC late to this paper.